# Inducible cell-to-cell signaling for tunable dynamics in microbial communities

Arianna Miano [1,2], Michael J. Liao [1,2] & Jeff Hasty [1,2,3 ✉]

The last decade has seen bacteria at the forefront of biotechnological innovation, with applications including biomolecular computing, living therapeutics, microbiome engineering and microbial factories. These emerging applications are all united by the need to precisely control complex microbial dynamics in spatially extended environments, requiring tools that can bridge the gap between intracellular and population-level coordination. To address this need, we engineer an inducible quorum sensing system which enables precise tunability of bacterial dynamics both at the population and community level. As a proof-of-principle, we demonstrate the advantages of this system when genetically equipped for cargo delivery. In addition, we exploit the absence of cross-talk with respect to the majority of well-characterized quorum sensing systems to demonstrate inducibility of multi-strain communities. More broadly, this work highlights the unexplored potential of remotely inducible quorum sensing systems which, coupled to any gene of interest, may facilitate the translation of circuit designs into applications.

[1] Department of Bioengineering, University of California San Diego, La Jolla, CA, USA. [2] BioCircuits Institute, University of California San Diego, La Jolla, CA, USA. [3] Molecular Biology Section, Division of Biological Science, University of California San Diego, La Jolla, CA, USA. ✉email: jhasty@ucsd.edu

Synthetic biology has the potential to revolutionize both healthcare and industry, with applications ranging from therapeutics[1–4] and drug delivery[5–7], to bioproduction[8] and bioremediation[9]. These emerging applications have uncovered the need to engineer spatially extended complex multi-cellular populations, requiring new tools that can bridge the gap between single cell, population, and community level engineering[10–13]. To achieve this, significant research efforts have been focused on engineering and characterizing a variety of cell-to-cell communication systems, with a particular focus on bacterial quorum sensing[14–17]. Currently, the majority of quorum sensing systems used in synthetic biology rely on self-produced small molecules that result in spatially and temporally self-organized systems, which can not be easily externally regulated[15,18–21]. In this study, we propose a tool that combines two pillars of population control: inducibility and cell-to-cell communication. To design this inducible quorum sensing system (iQS), we took inspiration from the native components of the photosynthetic bacterium *Rhodopseudomonas palustris*, which relies on a plant derived organic compound for the production of its signaling molecule[22]. This inducer, p-coumaric acid (pCA), is a ubiquitous molecule present in most fruits and vegetables[23] and has proven to be safe for both bacteria[24,25] and human cells[26,27] at relevant concentrations.

The iQS can be coupled with any gene of interest to enable tunable population density-dependent gene expression. We first demonstrated this principle by coupling the iQS system to the production of a fluorescent reporter protein in order to characterize the inducible circuit dynamics. Next, as a proof of concept, we coupled the iQS to a lysis gene, creating a tunable platform for cargo release. In direct comparison to non-inducible quorum sensing system, we demonstrate that the iQS significantly expands the range of population dynamics, allowing for temporal and spatial control of cargo release and population death. Finally, we exploit the orthogonality properties of the iQS system to demonstrate the ability to scale up inducibility from the population to the community level. In fact, it has been shown that the quorum sensing molecule produced by the bacterium *R. palustris* is orthogonal to the majority of well-characterized quorum sensing systems (Lux, Las, Tra, Rhl, Cin), providing a communication channel that can propagate information with minimal signal interference[16,17]. By co-culturing a two strain community, we demonstrate the ability to control population composition and dynamics by varying inducer concentrations. Overall, the iQS system combines many desirable characteristics into a single genetic circuit: inducibility, tunability, population-level coordination, inducer safety and orthogonality.

## Results

**Design and characterization of the iQS system**. To design the iQS system, we genetically reconstructed the two-step pathway, which converts pCA into p-coumaroyl-HSL (pC-HSL) through the production of the intermediate molecule p-coumaroyl-CoA[28] (Fig. 1a) . The first conversion is catalyzed by the p-coumaric acid-CoA ligase encoded by the *4CL2nt* gene from the plant *Nicotiana tabacum*[29] while the second step is catalyzed by the RpaI synthase. Additionally, we added superfolder green fluorescent protein (sfGFP) to be able to monitor the inducible dynamics. All genes, with the exception of *4CL2nt*, are driven by the pLux promoter, which has been shown to perform better than the native promoter of *R. palustris* when heterologously expressed[16]. We chose *Escherichia Coli* as the host chassis for our double plasmid circuit (Supplementary Table 1). We predicted that inducibility would expand the range of population dynamics by including an "OFF" state at low inducer concentration and an "ON" state at high inducer concentrations, which are both

independent of population density. On the other hand, for a range of intermediate concentrations the iQS would behave like a standard QS system by exhibiting the typical population density-dependent activation (Fig. 1b). To investigate how the engineered iQS strain would relate to these expected dynamics, we used microfluidic devices, which enabled the simultaneous exposure of varying pCA concentrations to different subgroups of cells. Time-lapse fluorescence microscopy was used to observe fluorescence expression as a function of population size and inducer concentration. The data matched the expected dynamics of an inducible quorum sensing system when exposed to zero, medium (15 nM) or high (1 μM) inducer concentrations (Fig. 1c). Additionally, we further characterized the iQS response over a broader range of pCA concentrations using a microwell plate-reader. Zero inducer resulted in the circuit being OFF except for a baseline of leaky expression. Intermediate levels (1 nM and 10 nM) induced the typical population-dependent switch-like behavior with a steep increase in fluorescence signal at an optical density (OD) threshold of around 0.3. Finally, high concentrations (100 nM to 10 μM) resulted in a linear relationship between fluorescence expression and population density, confirming the assumption of a population independent ON state (Fig. 1d). As a control, we repeated the same experiment with a wild-type *E. Coli* strain and observed that the growth curves were unaffected, confirming pCA is not toxic within the characterized range[24,25] (Fig. 1e).

**Characterization of the inducible syncronized lysis circuit (iSLC) in microfluidics**. To demonstrate the usefulness of the iQS over non-inducible quorum sensing-based circuits, we coupled the iQS with the expression of a lysis gene[5,7,30,31] to create a platform for inducible population dependant bacterial cargo delivery (Fig. 2a). To qualitatively predict the dynamics of this iSLC strain (induced synchronized lysis circuit), we developed a deterministic model (see details in the section "Methods"). A bifurcation plot was obtained by simulating the steady state values of cell population as a function of pCA concentration, predicting the emergence of three main population regimes (Fig. 2b). Small amplitude lysis events followed by steady growth are predicted at low inducer concentrations. Intermediate pCA concentrations result in sustained oscillations in population density. Finally, high inducer concentrations lead to a single lysis event with zero survivors. Therefore, we expected the iSLC to considerably expand the range of possible population dynamics by adding a quiescent state of circuit inactivation and a termination state in which all cells undergo lysis, regardless of population density (Fig. 2c).

A preliminary test using a microwell plate-reader showed an inverse correlation between the population OD at lysis and the inducer concentration (Supplementary Fig. 1). Interestingly, we noticed the presence of a lysis event at zero pCA concentration, which we proved to be caused by leaky expression of the pLux promoter in the sole presence of RpaR (Supplementary Fig. 2). To fully visualize the dynamics of the iSLC strain over time, we used microfluidic devices with an upstream serpentine of branching channels to generate a gradient of eight different inducer concentrations (Fig. 2d and Supplementary Fig. 3).

Using fluorescence microscopy to monitor population dynamics, we observed the same three emerging population behaviors predicted by the mathematical model (Fig. 2e, f and Supplementary Movies 1 and 2). With zero inducer, we observed constant cell growth with sporadic asynchronous lysis due to promoter leakiness. Intermediate inducer concentrations (15 nM) resulted in sustained synchronized oscillations with an average period of ~200 min. Finally, high pCA concentrations (110 nM to 1 μM) caused universal cell lysis regardless of population density,

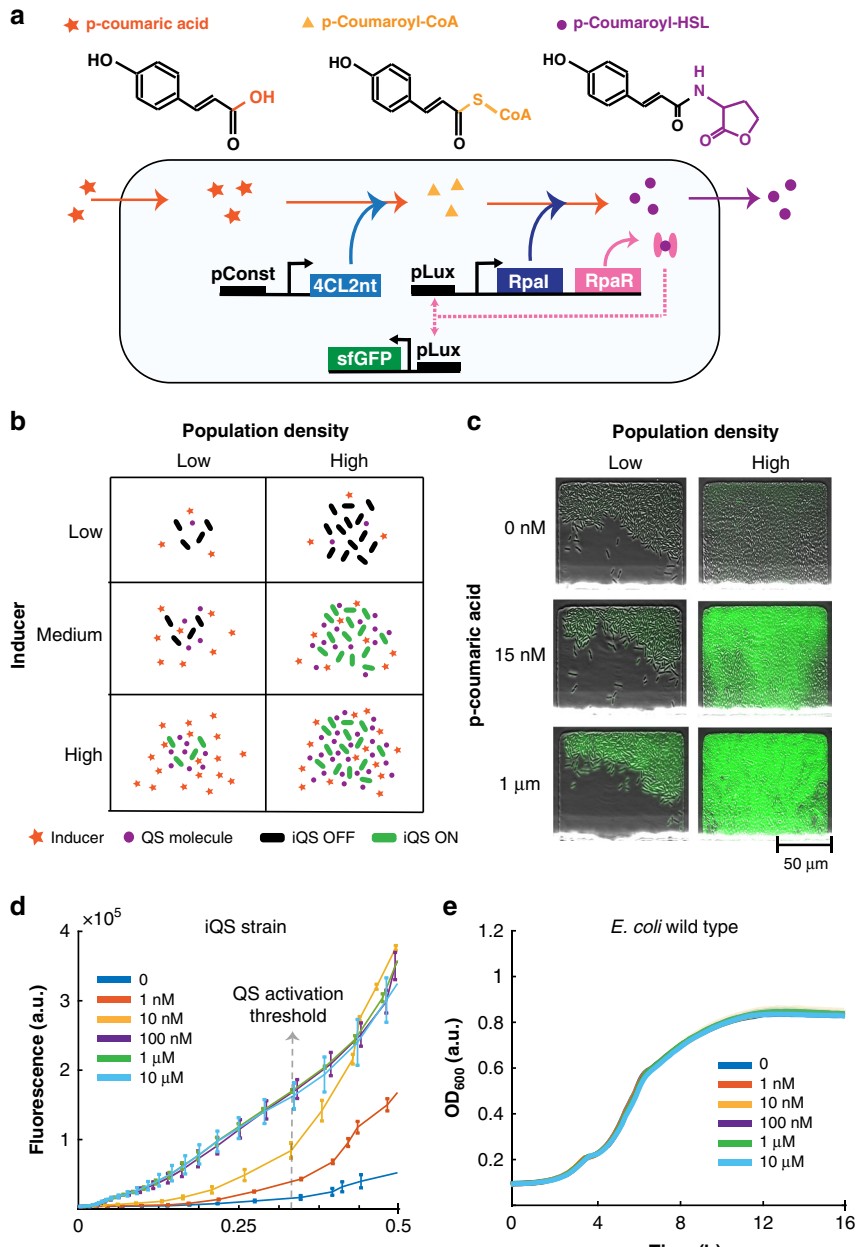

**Fig. 1 Design and characterization of the p-coumaric acid mediated iQS strain. a** Diagram of the iQS genetic circuit. The chemical structures of the molecules involved in the synthesis of the QS molecule are shown at the top. **b** Diagram to illustrate predicted dynamics associated with the inducible quorum sensing system as a function of population density and external inducer concentration. **c** Fluorescence microscopy images showing a composite of phase-contrast and GFP fluorescence in microfluidic traps. The data from the raw fluorescence values reflect the iQS dynamics predicted in part b. **d** Data from microplate reader experiment obtained by culturing the iQS strain in different p-coumaric acid concentrations. All data points represent mean ± standard deviation of three independent replicates. **e** Microplate reader experiment data obtained by culturing the wild-type *E. Coli* strain in a range p-coumaric acid concentrations. All data points represent mean (solid line) ± standard deviation of three independent replicates (shaded areas). Source data are provided as a Source Data file.

with only a small fraction (<3% of all microfluidics traps) able to survive. (Supplementary Fig. 4).

**Characterization of the iSLC kill switch.** Next, we hypothesized that modulating the inducer concentration in time would enable switching between states of constant growth (circuit quiescence), synchronized oscillations in population density (cyclic cargo release) and inducible population death (kill switch). To confirm this principle, we investigated the ability to drive the population dynamics from state one (population growth) to state three (population death) through pCA induction (Fig. 3a). We tested this hypothesis in microfluidic devices by initially growing the cells with zero inducer concentration (circuit quiescence). Following complete saturation of all traps, we induced with 500 nM pCA. We observed a synchronized lysis event throughout the entire device, which resulted in cell death in >97% of the 406 microfluidic traps present (Fig. 3b, c). To test the reproducibility of this property in larger volumes, we grew the iSLC strain in 3 ml culture tubes with and without inducer for a period of 8 h. In the

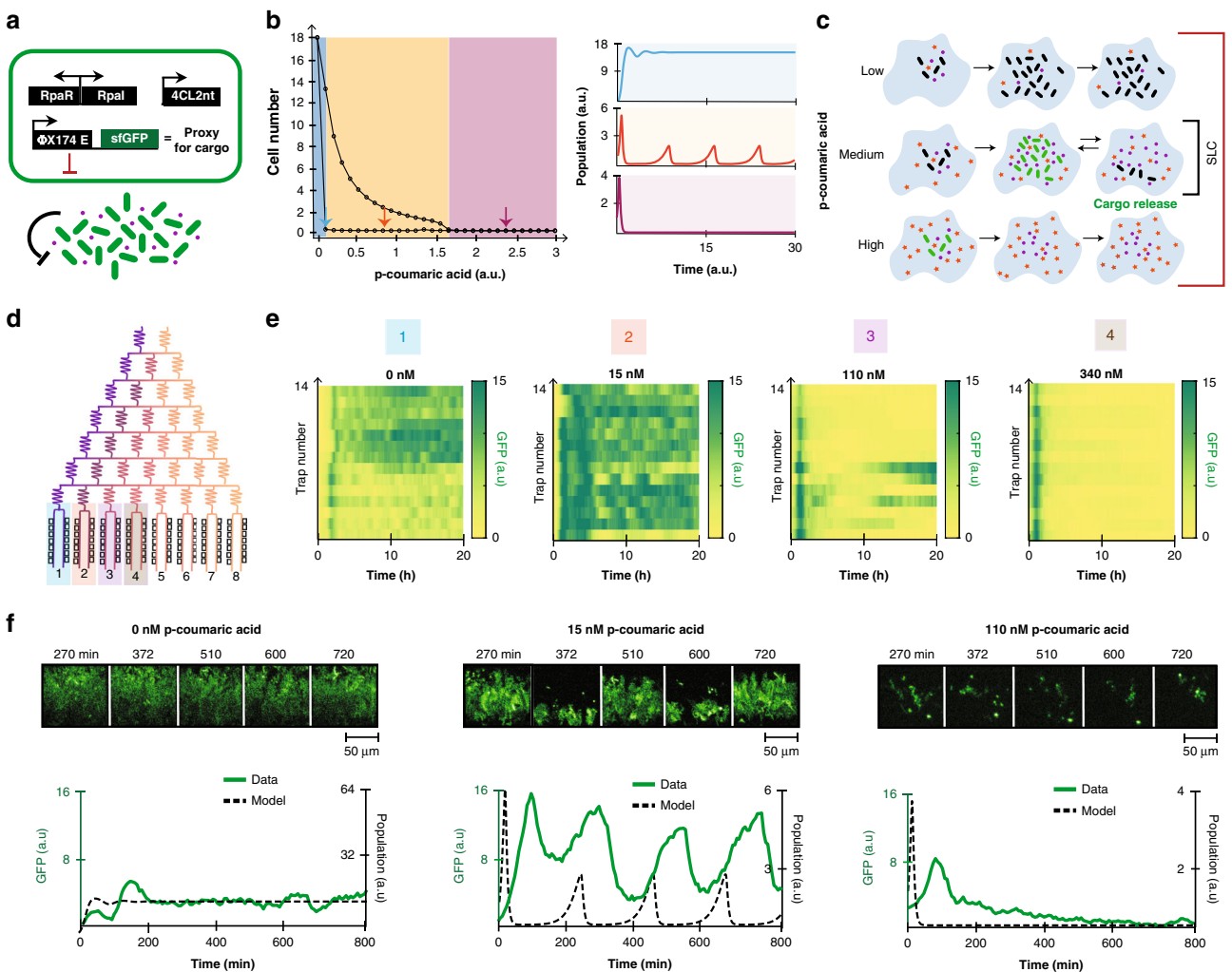

**Fig. 2 Characterization of the inducible syncronized lysis circuit (iSLC). a** Genetic diagram of the iSLC strain. **b** Simulations of the mathematical model showing three different dynamics at low (blue), medium (orange) and high (purple) inducer values. Medium values are predicted to result in sustained oscillations. Left: steady state maximum and minimum cell population values are plotted for a range of inducer concentrations. Right: simulated time traces for three representative p-coumaric acid values predict three emergent population dynamics. **c** Comparison between the iSLC and SLC dynamics based upon the model simulations in part b. **d** Diagram of the microfluidic device used to generate the inducer gradient. **e** Heatmaps representing the fluorescence time traces of all 14 traps present per column of the device. GFP signal is used as a proxy for population density. Four different inducer conditions are shown: low, medium, high, extra high, respectively. **f** Top: representative time series images from the fluorescence channel with three different inducer concentrations: low, medium, high, respectively. Bottom: fluorescence time traces plotted together with computer simulations of the mathematical model. For the simulation (dashed lines) time units are arbitrary, therefore the correspondence is strictly qualitative. Source data are provided as a Source Data file.

presence of p-coumaric acid, we observed a several order of magnitude decrease in cell survival, which was independent of inducer concentration (Fig. 3d and Supplementary Movie 3). This ability to bypass population dependency and intentionally decimate the population, regardless of its density, may serve as an integrated kill switch to regulate strain removal in space and time.

**Characterization of tunable multi-strain dynamics.** Finally, we investigated the orthogonal properties[16,17] of the iQS circuit by co-culturing the iSLC with a non-inducible SLC strain based on the Lux QS system, one of the most commonly used and well understood QS systems in synthetic biology[19,32] (Fig. 4a). First, we demonstrated the absence of cross-talk between p-coumaric acid and the Lux QS system by growing the iSLC strain in a range of different pCA concentrations (Supplementary Fig. 5). Next, we used the previously described microfluidic chip (Fig. 2d and

Supplementary Fig. 3) to co-culture the iSLC and SLC strains (starting at 1:1 ratio) under a gradient of inducer concentrations. Interestingly, we observed that by varying pCA concentration, we could precisely control the community composition of the microfluidic traps. Without inducer, the quiescent iSLC strain was able to displace the SLC strain due to an advantage in population growth rate. At intermediate concentrations (15 nM), we were able to maintain stable co-culture of both strains with independently sustained oscillations for >40 h. High pCA concentrations (110 nM) resulted in the death of all iSLC cells, allowing the SLC strain to fully takeover (Fig. 4b, c and Supplementary Movie 4). For inducer values higher than 340 nM, the dynamics were unanimously characterized by iSLC death and SLC takeover (Supplementary Fig. 6). As demonstrated, the absence of cross-talk between the pCA derived signaling molecule and the majority of well-characterized quorum sensing systems, enables the simultaneous use of multiple quorum sensing in

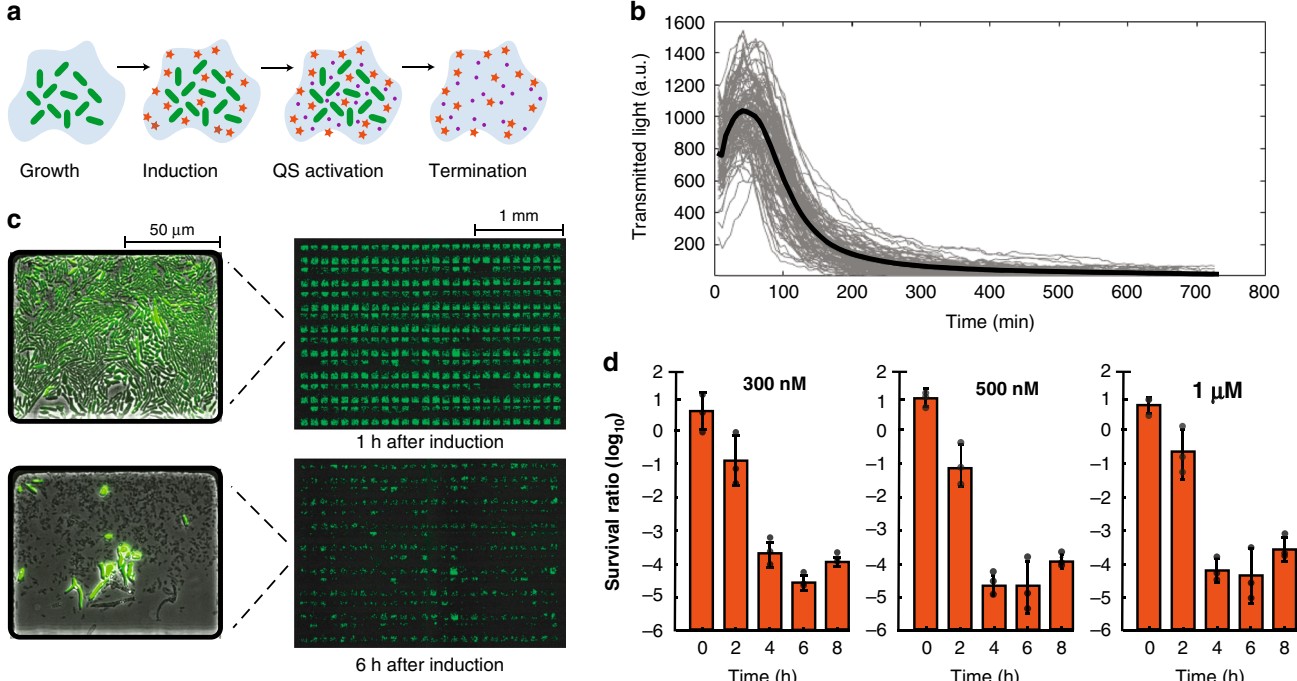

**Fig. 3 Characterization of the iSLC kill switch properties in microfluidics and liquid culture. a** Illustrated iSLC strain kill switch mechanism. **b** Example time traces ($n = 104$) extracted from the transmitted light channel (gray). Solid black line represent the mean. At time zero the cells were induced with 500 nM p-coumaric acid. **c** Movie stills (×4) of the microfluidic chip before (top) and 6 h after (bottom) induction with 500 nM p-coumaric acid. Left side shows magnified images (×30) of a single representative trap. **d** Cell viability measured by CFU count following addition of the killing signal (p-coumaric acid) in liquid culture. Individual data points are represented by circles, the bars represent mean ± standard deviation of the three independent replicates. Source data are provided as a Source Data file.

co-culture, providing exciting new possibilities for microbial community engineering.

## Discussion

Given the importance of dynamic gene expression in nature[33,34] and the increasing availability of tools for modular and robust design of genetic circuits[12,17], synthetic biologists have been attracted towards systems that can achieve tunable complex behaviors, such as the iQS, as opposed to simple steady state dynamics. Our results demonstrate the broad potential of the iQS platform to enable a wide range of new functionality for synthetic circuits that rely on cell-to-cell communication systems for population-level coordination. In particular, we show how inducibility broadly expands the functionality of quorum sensing-based circuits by enabling switching between states of circuit quiescence, quorum-dependent circuit activation and population wide constitutive expression. As a proof-of-principle, we created the iSLC, which enables spatial and temporal modulation of inducer-dependent cargo release for single or multi-strain communities. In particular, we demonstrate the ability to temporally and spatially control transitions between inactivated population growth, quorum enabled cyclic cargo release and global population death (kill switch). This precise control of circuit activation, the ability for timed strain elimination and the non-toxic nature of p-coumaric acid make this system particularly attractive for potential therapeutic applications in vivo. Finally, we exploited the orthogonality of the iQS system to demonstrate precise control of multi-strain dynamics, which has potential to become a key tool for engineering synthetic bacterial communities. Although the circuit functionalities of the iQS were demonstrated in the *E. coli* strain MG1655, we believe that the circuit could potentially be extended to other bacterial species. We expect the main challenge to be the functional expression of the p-coumaric

acid-CoA ligase enzyme (encoded by the *4CL2nt* gene) due to its heterologous plant origins. Possible solutions to this challenge might include species-specific codon-optimization techniques or the use of homologous proteins.

Looking ahead, the modularity and simplicity of the iQS genetic parts make it straightforward for coupling to any gene of choice, enabling precise spatial and temporal control over population-level gene expression. This may open up new possibilities for sophisticated, and safe, biotechnological applications. Overall, this work demonstrates the translation of more than a decade of circuit design into microbiological organization, from molecular regulatory mechanisms (synthetic promoters), to single cell protein expression (enzyme catalysis), multi-cellular population coordination (cell-to-cell communication) and multi-species interaction (orthogonal quorum sensing).

## Methods

**Plasmids and strains.** Our iSLC and SLC strains were both cultured in lysogeny broth (LB) media with 50 μg ml[−1] kanamycin, 34 μg ml[−1] chloramphenicol and 0.2% glucose for strains containing ColE1 origin and p15A origin plasmids in a 37 °C shaking incubator. Plasmid pTD103 RpaR-RpaI-LAA-sfGFP[21], ptD103-CFP[19], and pZA35-X174E(+LuxR)[5] were constructed by our group in previous studies (Supplementary Fig. 7). Both plasmids pAM014 and pAM021 were obtained by inserting the gene *4CL2nt*[28] under the constitutive promoter J23106 from the Anderson promoter library. The *4CL2nt* gene and the promoter were synthesized with overlapping PCR of long oligos (IDT). All plasmids were constructed by Gibson assembly followed by transformation into DH5α (Thermofisher) chemically competent *E.coli*. All plasmids were verified by Sanger sequencing before transformation into *E.coli* strain MG1655.

**Microfluidics and microscopy.** The microscopy and microfluidics techniques used in this study are similar to those previously reported by our group[35]. Our microfluidic devices were constructed from PDMS (poly-dimethylsiloxane), which was molded and baked on a silicon wafer with micron-scale features formed by cross-linked photoresist. Once the PDMS hardened, it was peeled off, and individual devices were cut out. In order to connect fluid lines to the device, holes

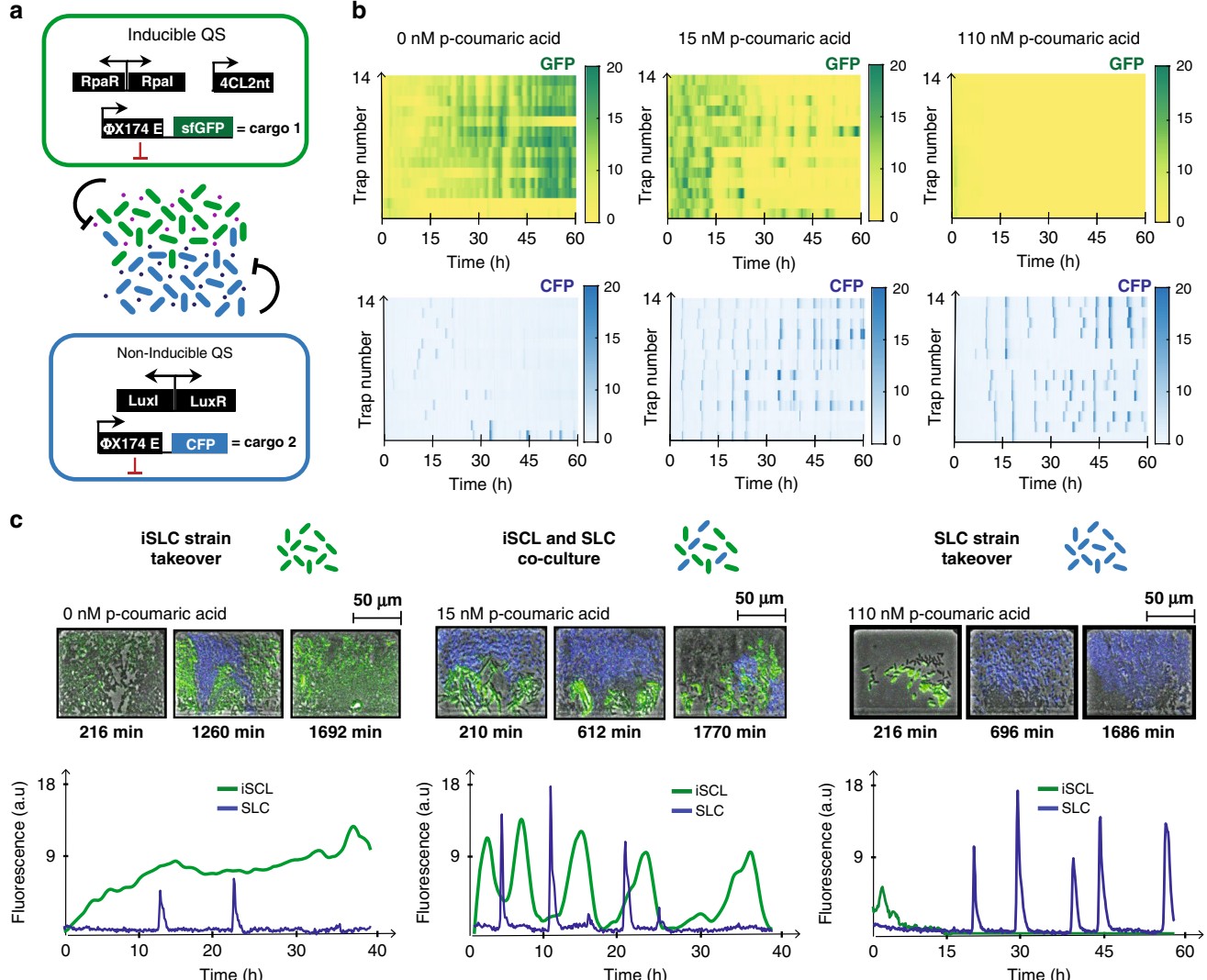

**Fig. 4 iQS enabled modulation of orthogonal multi-strain dynamics. a** Schematics representing co-culture of the two strains used with orthogonal inducible (iSLC) and non-inducible (SLC) quorum sensing, respectively. **b** Heatmaps representing the fluorescence time traces of all 14 traps present per column of the device. Top rows show the GFP values and bottom rows the CFP values. Fluorescence signals are used as a proxy for population density. **c** Shown at the top are Movie stills of the co-culture for three inducer concentrations at multiple time points. The corresponding fluorescence time traces for GFP and CFP are plotted at the bottom. Source data are provided as a Source Data file.

where punched in correspondence of the inlets and outlets of the device. Afterwards, the devices were left for 30 min in a vacuum chamber. Meanwhile, 1 ml of overnight cell culture was spun down by centrifugation and resuspended in 10 μl of fresh media with appropriate antibiotics. After taking the device out of the vacuum chamber, a single droplet of re-suspended cells was positioned in correspondence of the outlet opening. Similarly, droplets of sterile fresh media were placed in correspondence of the inlets openings[36]. In all cases, 0.075% Tween20 was added to the medium to prevent cells from sticking to the PDMS walls. After all chip features were wetted, the fluids lines were plugged in and the height of the inlet was raised 10 to 40 cm above the device. The outlet syringe was instead placed at the same height of the device. For co-culturing experiments in Fig. 4, cells were cultured individually overnight and eventually spun down and re-suspended together (1:1 ratio) allowing for a single droplet to be loaded on the device. All experiments shown in Figs. 2 and 4 were performed in a side-trap array device with bacteria growth chambers ~100 × 80 μm in area and ~1.2 μm in height. The upstream channels consists of a series of dividing serpentine branches, which allow for sequential dilutions of the two input media, generating a gradient of eight different inducer concentrations (Supplementary Fig. 3). On the other hand, for the kill switch experiments, reported in Fig. 3, we used a simpler device with a single input and an ordered array of traps[35]. The dimensions of the traps are the same as the ones described for the gradient device. Experiments in Figs. 2 and 4 where carried out by connecting a syringe with LB + antibiotics + 0.075% Tween20 as inlet 1 and a syringe with LB + antibiotics + 0.075% Tween20 + 1μM of p-coumaric acid as inlet 2. P-coumaric acid inductions for microfluidic experiments in Fig. 3b were performed by unplugging the syringe with pure media and substituting it with a second syringe containing media plus the appropriate acid concentration. For microscopy we used the same system as described in our previous work[35]. In brief, images were acquired with a Nikon TI2 using a Photometrics CoolSnap cooled charge-coupled device (CCD) camera. The scope and accessories were programmed using the Nikon Elements software. The microscope was housed in a plexiglass incubation chamber maintained at 37 °C by a heating unit. Phase-contrast images were taken at x4 and x10 magnification at 50–100 μs exposure times. At x4 magnification fluorescence exposure times were 2 s at 30% intensity for both gfp and cfp while at x10 magnification they were 200 μs at 30% intensity for both gfp and cfp. Images were taken every 6 min for each experiment. For induction experiments, imaging was paused while syringes were swapped.

**Data analysis**. Fluorescence intensity profiles were obtained by analyzing frames from the fluorescent channels. The mean fluorescence values were calculated by drawing a rectangle surrounding each trap individually and extracting the z-axis profile on ImageJ. Fluorescence values shown in Fig. 2 were normalized by dividing all data by a constant factor. In addition, the subset of traps for each column was normalized by subtracting the minimum value among the traps within the subset. Transmitted light data from Fig. 3b was normalized by subtracting the minimum value for each time trace. Fluorescence data shown in Fig. 4 was normalized by first dividing the subsets of traps by their overall maximum values and subsequently subtracting the respective minimum values. In addition, the data in Fig. 4 was

smoothed with the command *smoothdata()* in Matlab. Heatmaps were generated in Matlab using the function *heatmap()*. For the co-culture experiments, when overlap between the gfp and cfp channels was observed, values were corrected taking into consideration the image frames in order to subtract overlapping signal.

**Plate-reader experiments**. For plate-reader experiments, the appropriate strains were seeded from a $-80\,°C$ glycerol stock into 3 ml LB with 0.2% glucose and appropriate antibiotics and incubated in a 37 °C shaking incubator. The following day, 2 μl of overnight culture were added to 200 μl of fresh media with appropriate antibiotics in a standard Falcon tissue culture 96-well flat bottom plate. Cells were incubated at 37 °C shaking in a Tecan Infinite M200 Pro. Cells were grown for about 12 h. The OD at 600 nm absorbance was measured every 10 min.

**Cells survival assay**. Cell viability assay to test the efficacy of iQS as a kill switch was done measuring colony forming units (CFUs), following a protocol found in the literature[37]. Cells were grown under survival conditions in LB with 0.2% glucose, which inhibits the LuxI promoter thanks to the presence of a binding site for the CAP-cAMP activating complex[38]. In the morning, they were transferred into four liquid cultures of fresh LB medium with 0.2% glucose, 300 nM p-coumaric acid, 500 nM p-coumaric acid and 1 μM p-coumaric acid, respectively. Samples were collected every 2 h and serially diluted in PBS over a 7-log range and spotted (2 μl) onto LB agar plates with 0.2% glucose. The equations used are: CFU/ml = (number of colonies) × (dilution factor)/0.002 mL, survival ratio ($log_{10}$) = log (CFU/ml with glucose)/(CFU/ml with p-coumaric acid).

**Modeling**. We constructed a deterministic model to qualitatively describe the dynamic behavior of the iSLC strain. The model is based on a set of six ordinary differential equations (ODEs), which track the evolution of the following six variables: the cell number into a single microfluidic trap ($N$), the external concentration of p-coumaroyl-HSL ($pH$), the intracellular concentration of lysis protein ($L$), the intracellular concentration of the pC-HSL synthase RpaI ($I$), the intracellular concentration of p-coumaric acid-CoA ligase ($E$) and the intracellular concentration of the intermediate compound p-coumaroyl-CoA ($pA$). The inducer concentration is defined as a fixed parameter.

A non-zero p-coumaric acid concentration induces the production of the intermediate molecule p-coumaroyl-CoA ($pA$) through the conversion mediated by the p-coumaric acid-CoA ligase (encoded by gene *4CL2nt*) ($E$). The production term of the latter is defined as a constant variable according to the constitutive promoter, which drives it. The intermediate product ($pA$) is eventually transformed into the quorum sensing molecule pC-HSL through the RpaI synthase enzyme ($I$). Once a threshold value of extracellular pC-HSL is reached, the intracellular production of the pLux promoter driven genes (RpaI, RpaR, and E) are brought to the ON state. The same promoter also drives the lysis gene, therefore positive feedback also results in cell lysis. We assume that the quorum sensing molecule diffuses quickly through the membrane, therefore we do not distinguish between intracellular and extracellular HSL concentration. In addition, we assume that the pC-HSL-RpaR complex binding is instantaneous, so that the model can be simplified by ignoring the dynamics of the binding complex. Degradation of all proteins ($L$, $I$, $E$) is associated with dilution due to cell growth ($\mu_G$), as well as basal intracellular degradation ($\gamma_L$ and $\gamma_I$). In addition to those terms, RpaI ($I$) is also actively degraded by ClpXP proteases ($\gamma_C$). Overall, our model can accurately predict the three main dynamics of the bacterial population as the inducer concentration is varied. With zero inducer concentration, the population grows reaching a steady state value. Similarly, at very small inducer concentrations, the population undergoes small amplitude lysis events followed by steady state. We observed a finite range of intermediate inducer values, which resulted in sustained oscillations of population density. Finally, we observed total population death with no survivors for high concentrations (Supplementary Fig. 8). We can visualize the non-linear dynamics of this system using phase portraits obtained by plotting N (cell number) against L (lysis protein) (Supplementary Fig. 9). As the p-coumaric acid concentration is increased, the simulations show a first transition from a stable spiral to a limit cycle, which indicates sustained oscillations. A further increase in the inducer parameter causes the limit cycle to disappear in favor of a stable fixed point. All plots were generated in MATLAB.

$$\frac{dN}{dt} = \mu_G * N * (N_0 - N) - N * \frac{k * L^n}{(L_0)^n + L^n} \quad (1)$$

$$\frac{dpA}{dt} = \mu_4 * \text{inducer} * E - \gamma_{CoA} * pA - \mu_G * pA \quad (2)$$

$$\frac{dpH}{dt} = \mu_H * N * I * pA - \frac{u * pH}{1 + \frac{N}{N_0}} \quad (3)$$

$$\frac{dL}{dt} = C_l * \left( \alpha_0 + \frac{\alpha_H * \left(\frac{pH}{H_0}\right)^4}{1 + \left(\frac{pH}{H_0}\right)^4} \right) - \gamma_L * L - \mu_G * L \quad (4)$$

$$\frac{dI}{dt} = C_i * \left( \alpha_0 + \frac{\alpha_H * \left(\frac{pH}{H_0}\right)^4}{1 + \left(\frac{pH}{H_0}\right)^4} \right) - \gamma_I * I - \mu_G * I \quad (5)$$

$$\frac{dE}{dt} = C_l * P_{const} - \gamma_4 * E - \mu_G * E - \gamma_C * E \quad (6)$$

We chose model parameters based on a similar model previously published[5]. Compared to the Lux quorum sensing system, the iSLC showed higher promoter leakiness, which was taken into account by increasing the basal production term ($\alpha_0$). The parameter values used in the model are $\mu_G = 0.2$ (dilution due to cell growth), $N_0 = 20$ (cell capacity of a single trap), $k = 10$ (maximum rate of cell lysis), $L_0 = 1$ (concentration of lysis protein resulting in half maximum lysis), $n = 2$ (Hill's coefficient), $C_l = 0.5$ (copy number of the lysis gene), $C_i = 1$ (RpaI gene copy number), $P_{const} = 20$ (strength constitutive promoter driving gene *4CL2nt*), $\gamma_4 = 2$ (degradation of enzyme p-coumaric acid-CoA ligase), $\alpha_0 = 1$ (pLux basal leakiness), $\alpha_H = 20$ (Lux promoter pC-HSL-RpaR induced production), $H_0 = 4.5$ (pC-HSL-RpaR binding affinity to pLux), $\gamma_L = 2$ (lysis protein basal degradation), $\gamma_I = 2$ (RpaI protein basal degradation), $\gamma_C = 12$ (RpaI protein degradation due to ClpXP), $\gamma_{CoA} = 2$ (p-coumaroyl-CoA basal degradation), $\mu_4 = 30$ (conversion rate of p-coumaric acid into p-coumaroyl-CoA), $\mu_H = 15$ (pC-HSL production rate) and $\mu = 12$ (maximum AHL clearance rate due to flow).

**Reporting summary**. Further information on research design is available in the Nature Research Reporting Summary linked to this article.

## Data availability
Authors can confirm that all relevant data are included in the paper and/or its supplementary information files. The source data underlying Figs. 1d–e, 2e, 3b, d and 4b are provided as a Source Data file. All other data, plasmids and bacterial strains are available from the corresponding author upon request.

## Code availability
All code is available from the authors upon request.

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

## Acknowledgements

We would like to thank M. Omar Din for the stimulating conversations, Rachel Ng for the micro-fabrication of the wafer, Philip Bittihn for providing the original CAD design of the gradient chip, Aida Martín and Andrew Lezia for critical reading of the manuscript. This work was supported by the NSF (award MCB1616997).

## Author contributions

A.M., M.J.L., and J.H. contributed to the development of the project. A.M. constructed the plasmids and strains and analyzed results. A.M. and M.J.L. conducted the experiments. A.M. conducted the mathematical analysis and computational modeling. A.M. prepared the figures, and A.M., M.J.L., and J.H. prepared the manuscript.

## Competing interests

A patent application (U.S. provisional patent application no. 62/947,932) has been filed on the inducible signaling for tunable microbial dynamics. J.H. has a financial interest in GenCirq. The remaining authors declare no competing interests.
