## [Peer Review File · Nature Communications]

Reviewers' comments:

Reviewer #1 (Remarks to the Author):

The manuscript by Miano and colleagues describes a new synthetic gene circuit allowing to achieve population-level coordination of gene expression. The work builds on the "synchronised lysis circuit" the group published in the recent past. Notably, the iSLC circuit presented in this article adds "external control" to the previous gene network, rendering it a significant contribution to the Synthetic Biology community.

The manuscript is well structured and presented. The idea underlying the work is innovative, the science presented solid and the results extremely compelling. I am persuaded that this is an elegant and versatile piece of research that deserves to be published in a prestigious outlet. I only have minor comment on the presentation itself: I am wondering whether, to increase the impact of the findings (e.g. reaching out to a wider readership), the authors should consider reframing the inducibility as a tunability. From a systems perspective, the concentration of p-coumaric acid can be thought of as a "parameter" rather than an input; by modulating the concentration of this compound we are effectively "morphing" the circuit.

Typos:

p.6 "Modelling" ... intermedlate

p.7 second line after the equations: "publishedcitedin"

Reviewer #2 (Remarks to the Author):

In this manuscript, Miano et al presents a novel inducible quorum sensing (iQS) system that is based on p-coumaric acid and enables precise control over bacterial dynamics both at the population and community level. They first illustrate the system by coupling it to the production of a fluorescent reporter protein in order to characterize the inducible circuit dynamics. Next, they couple the iQS to a lysis gene which leads to a tunable platform for cargo release. In comparison to non-inducible quorum sensing system, the authors further demonstrate that the iQS significantly expands the range of population dynamics, allowing for temporal and spatial control of cargo release and population death. They finally show the orthogonality properties of the iQS system to scale up inducibility from the population to the community level.

Together, this study demonstrates the four unique characteristics of the iQS system—inducibility, population-level coordination, inducer safety and orthogonality—which offers broad and versatile applications in a variety of settings when coupled to other gene circuits. It thus offers a new dimension of controllability for gene circuits, which is urgently needed for the field of synthetic biology, and thus enriches the gene circuit arsenals of synthetic biology particularly for the design and control of microbial communities. The paper has convincing results and a seamless integration of experiment with modeling. It is also well organized and clearly presented. I thus warmly recommend publications with the following minor comments being addressed.

1. Figure 3: The authors reported that the synchronized lysis event throughout the entire device resulted in ~3% of surviving cells. However, in larger volumes, it showed more than 4 orders of magnitude reduction for the CFU counting for the uninduced and induced cases. It thus seems that there are 2 orders of magnitude difference of the survival in the two settings. The authors need to justify the origin of the significant difference.

2. The utility of the p-coumaric acid induced system was demonstrated in *E. coli*. However, it could in principle be extended to other microbial species. Please discuss the possibility of extension and potential limitations of the platform when applied to different microbial species (e.g., gram positive species).

Responses to Referee Critiques

General Comments to the referees

We are grateful for their appreciation of the quality and relevance of our study in terms of impact. The specific suggestions were valuable in guiding further editing of the manuscript which have strengthened numerous aspects of the work.

Response to Referee #1

The manuscript by Miano and colleagues describes a new synthetic gene circuit allowing to achieve population-level coordination of gene expression. The work builds on the "synchronised lysis circuit" the group published in the recent past. Notably, the iSLC circuit presented in this article adds "external control" to the previous gene network, rendering it a significant contribution to the Synthetic Biology community.

The manuscript is well structured and presented. The idea underlying the work is innovative, the science presented solid and the results extremely compelling. I am persuaded that this is an elegant and versatile piece of research that deserves to be published in a prestigious outlet. I only have minor comment on the presentation itself: I am wondering whether, to increase the impact of the findings (e.g. reaching out to a wider readership), the authors should consider reframing the inducibility as a tunability. From a systems perspective, the concentration of p-coumaric acid can be thought of as a "parameter" rather than an input; by modulating the concentration of this compound we are effectively "morphing" the circuit.

We thank the referee for their high opinion of our work and the useful suggestion to improve the impact of our findings. We have now edited the manuscript in the abstract, introduction and discussion in order to emphasize the concept of tunability when

referring to the characteristics of the iQS system. The sentences we edited are listed below:

To address this need, we engineer an inducible quorum sensing system which enables precise tunability of bacterial dynamics both at the population and community level.

The iQS can be coupled with any gene of interest to enable tunable population density-dependent gene expression.

Overall, the iQS system combines many desirable characteristics into a single genetic circuit: inducibility, tunability, population-level coordination, inducer safety and orthogonality.

Given the importance of dynamic gene expression in nature and the increasing availability of tools for modular and robust design of genetic circuits, synthetic biologists have been attracted towards systems that can achieve tunable complex behaviors, such as the iQS, as opposed to simple steady state dynamics.

Typos:

p.6 “Modelling” ... intermediate.

p.7 second line after the equations: “publishedcitedin”.

We thank the referee for pointing out these typos which have been corrected in the manuscript.

Response to Referee #2

In this manuscript, Miano et al presents a novel inducible quorum sensing (iQS) system that is based on p-coumaric acid and enables precise control over bacterial dynamics both at the population and community level. They first illustrate the system by coupling it to the production of a fluorescent reporter protein in order to characterize the inducible circuit dynamics. Next, they couple the iQS to a lysis gene which leads to a tunable platform for cargo release. In comparison to non-inducible quorum sensing system, the authors further demonstrate that the iQS significantly expands the range of population dynamics, allowing for temporal and spatial control of cargo release and population death. They finally show the orthogonality properties of the iQS system to scale up inducibility from the population to the community level.

Together, this study demonstrates the four unique characteristics of the iQS system—inducibility, population-level coordination, inducer safety and orthogonality—which offers broad and versatile applications in a variety of settings when coupled to other gene circuits. It thus offers a new dimension of controllability for gene circuits, which is urgently needed for the field of synthetic biology, and thus enriches the gene circuit arsenals of synthetic biology particularly for the design and control of microbial communities. The paper has convincing results and a seamless integration of experiment with modeling. It is also well organized and clearly presented. I thus warmly recommend publications with the following minor comments being addressed.

We thank the referee for their strong support of the manuscript, their attention to detail, along with the suggestions below that have improved the quality of the study.

1. **Figure 3:** The authors reported that the synchronized lysis event throughout the entire device resulted in ~3% of surviving cells. However, in larger volumes, it showed more than 4 orders of magnitude reduction for the CFU counting for the uninduced and induced cases. It thus seems that there are 2 orders of magnitude

difference of the survival in the two settings. The authors need to justify the origin of the significant difference.

We thank the referee for pointing out that confusion, we should have been clearer here. The 3% survival rate associated to the microfluidic experiments refers to the percentage of microfluidics traps in which we could detect surviving cells. In these scenarios, the recolonization of the trap could be the result of just a single cell surviving the death event. Therefore, this was a very qualitative characterization of the killing properties of the iSLC circuit which did not rely on direct cell count.

While it would be possible to estimate the survivor rate in microfluidics (under the assumption that each trap holds roughly 5,000 cells), we felt that a more quantitative representation could be achieved by repeating the experiment in liquid culture where we could accurately perform CFU counting.

2. The utility of the p-coumaric acid induced system was demonstrated in E. coli. However, it could in principle be extended to other microbial species. Please discuss the possibility of extension and potential limitations of the platform when applied to different microbial species (e.g., gram positive species).

We thank the referee for this suggestion and agree that such a discussion will be a valuable addition to the manuscript. We have appended the manuscript with the following text:

Although the circuit functionalities of the iQS were demonstrated in the E.Coli strain MG1655, we believe that the circuit could potentially be extended to other bacterial species. We expect the main challenge to be the functional expression of the p-coumaric acid-CoA ligase enzyme (4CL2nt) due to its heterologous plant origins. Possible solutions to this challenge might include species specific codon-optimization techniques or the use of homologous proteins.

REVIEWERS' COMMENTS:

Reviewer #2 (Remarks to the Author):

I think the authors have addressed the comments from me and the other reviewer. I warmly recommend publication.